# Position: The Inevitable Transition to Machine Learning in Quantum Chemistry

**Karen Sargsyan** [1]   **Chao-Ping Hsu** [1]

## Abstract

Finding exact solutions to the quantum many-body problem is computationally intractable (QMA-hard). Traditional approximations for electrons in an atom or molecule—density functional theory and wavefunction methods—have been indispensable, but their development shows signs of saturation: DFT functionals have proliferated without converging toward the exact functional, and strong correlation remains largely unsolved after decades of effort. This position paper argues that machine learning represents the most promising path forward—not as a proof of logical necessity, but as a decision-theoretic argument: ML succeeds whether the underlying problems are truly hard or merely lack simple analytical solutions. We reframe recent traditional method development as "hand-crafted machine learning" that has exhausted the hypothesis space accessible to human intuition. Significant challenges remain, but these have clear research paths forward, unlike the fundamental barriers facing traditional approaches. ML-based approaches merit strategic priority in quantum chemistry's next phase.

## 1. Introduction

**Position: Machine learning should be the primary direction for developing new quantum chemistry approximations. Traditional methods remain valuable as everyday computational tools and as generators of reference data, but the hand-crafted development of new functionals and wavefunction ansatze has reached diminishing returns.** This is a claim about strategic research priority, not a call to displace traditional methods from practice. We develop the position through five interlocking claims (Positions 1–5), beginning with the computational-complexity foundations in the next section.

[1]Institute of Chemistry, Academia Sinica, Taipei, Taiwan. Correspondence to: Karen Sargsyan <karen.sarkisyan@gmail.com>.

*Proceedings of the 43rd International Conference on Machine Learning*, Seoul, South Korea. PMLR 306, 2026. Copyright 2026 by the author(s).

## 2. The Intractable Quantum Challenge

Computational quantum chemistry is foundational to materials science(Marzari et al., 2021; He et al., 2019), our understanding of complex biological processes(Kircheva et al., 2025; Dudev & Lim, 2014; Borges et al., 2021), rational drug design(Mazmanian et al., 2022; Kircheva et al., 2024; Cavalli et al., 2006; Niazi, 2025), and addressing major challenges facing humanity(Sandoval-Pauker et al., 2024; Siahrostami & Murray, 2024). The central ambition is to solve the time-independent Schrödinger equation:

$$\hat{H}\Psi = E\Psi \tag{1}$$

where $\hat{H}$ is the Hamiltonian operator, $E$ is the energy, and $\Psi$ is the wavefunction.

However, the many-electron wavefunction $\Psi$ exists in a Hilbert space whose dimension scales exponentially with the number of electrons, $N$. This "curse of dimensionality" renders direct solutions intractable for systems beyond a few atoms.

### 2.1. The Limits of Hardware and Universal Algorithms

A common misconception is that continued growth of computational power will eventually overcome this challenge. This view misunderstands the nature of the complexity. Traditional high-accuracy methods, such as Full Configuration Interaction (FCI), scale factorially ($O(N!)$)(Gao et al., 2024; Helgaker et al., 2013). Even sophisticated approximations like CCSD(T) scale as $O(N^7)$(Helgaker et al., 2013). Exponential or high-order polynomial complexity cannot be defeated by polynomial improvements in hardware.

This practical difficulty is formalized by computational complexity theory(Arora & Barak, 2009). Finding the ground state energy of a general quantum system is QMA-complete(Kitaev et al., 2002). QMA (Quantum Merlin-Arthur) is the quantum analogue of the classical complexity class NP. The hardness pervades multiple formulations: the N-representability problem—determining whether a two-body density matrix derives from a valid N-particle state—is also QMA-complete(Liu et al., 2007), and finding the global Hartree-Fock minimum is NP-hard due to multiple local minima(Whitfield et al., 2013). These results imply that even fault-tolerant quantum computers will likely not

efficiently solve general electronic structure problems exactly.

Beyond efficiency, determining the spectral gap for arbitrary materials is provably undecidable(Cubitt et al., 2015), reinforcing the intractability of general quantum systems.

## 2.2. The Space-Time Tradeoff and Specialization

The intractability of the general problem necessitates a shift in strategy, underpinned by the Space-Time Tradeoff (TSTO)(Hellman, 1980). This principle from computer science dictates a fundamental exchange: computational time can often be reduced by increasing the use of memory (space), and vice-versa.

Traditional *ab initio* methods like FCI sit at one extreme of this tradeoff: they store no chemistry-specific knowledge and must recompute everything from first principles for every new system, with intractable time complexity as the price. The memory FCI does require—to hold the CI vector—is a computational intermediate, not the kind of precomputed advice the tradeoff concerns. Traditional quantum chemistry already trades space for time on smaller scales: basis sets, contracted integrals, and active-space selections are precomputed and reused rather than rederived every time. ML applies the same logic at far larger scale. The opposite extreme would be a complete lookup table containing precomputed answers for all possible chemical systems—requiring minimal computation time but infeasible storage due to the vastness of chemical space. If a universal, efficient algorithm is impossible (as suggested by complexity results), the only viable path to reduce computational time is to move along the tradeoff frontier by utilizing more space to store specialized knowledge.

This strategy is formalized in computer science by distinguishing between uniform computation—the search for a single algorithm that works for all inputs—and non-uniform computation, characterized by the complexity class P/poly. This framework recognizes that even if a universal solution is intractable, we might construct a collection of specialized solutions. P/poly describes problems solvable by a family of efficient circuits supplemented by "advice"—pre-computed information tailored to specific domains. A trained neural network is precisely such a circuit: the training process discovers specialized knowledge, and the resulting network weights constitute the stored advice.

The quantitative implications of this tradeoff are striking. Drug-like chemical space alone is estimated to contain $10^{33}$–$10^{60}$ possible molecules(Bohacek et al., 1996; Reymond, 2015), yet modern machine learning potentials achieve useful accuracy with only $10^6$–$10^7$ parameters(Batatia et al., 2025). We use drug-like space because it is the best-quantified domain; the same compression principle ap-

plies to the broader spaces relevant to quantum chemistry—materials, catalysis, and extended systems—where the numerical estimates are less precise but the qualitative argument is identical. This compression ratio of $10^{26}$–$10^{53}$ is only possible because chemistry possesses exploitable structure—locality, smoothness, and physical symmetries—that learned representations can capture. The model does not memorize chemical space; it learns generalizable patterns that enable accurate interpolation and extrapolation. We provide a detailed quantitative analysis of this phenomenon in Appendix A.

## 2.3. The Decision-Theoretic Case for ML

A natural objection to our complexity-based argument is that QMA-hardness is a worst-case result for arbitrary Hamiltonians. Real chemistry occupies a tiny, structured subset—stable molecules with physical potentials obeying locality and smoothness constraints. Perhaps this subset possesses exploitable structure that renders it tractable without machine learning.

This objection cannot be dismissed on theoretical grounds alone. We do not know *a priori* whether chemically relevant problems inherit the hardness of the general case. However, the ML paradigm remains the rational choice under this uncertainty because it succeeds across a wide range of scenarios. If chemically relevant problems are QMA-hard or comparably difficult, traditional compact algorithms will fail and ML becomes necessary. If they are easier but still lack simple analytical solutions, ML provides an efficient path to discover specialized approximations. And if simple analytical solutions do exist, ML-based exploration of functional space may help identify them. The key asymmetry is that committing exclusively to traditional approaches fails catastrophically if the subset is hard, while ML-based approaches degrade gracefully.

The AlphaFold precedent offers a compelling parallel. Protein structure prediction—determining the three-dimensional shape of a protein from its amino acid sequence—had been a grand challenge in biology for over fifty years. The general problem of protein folding is NP-hard(Fraenkel, 1993), placing it in an analogous complexity class to QMA-hardness for quantum chemistry. For decades, researchers debated whether natural proteins—a structured subset constrained by evolutionary selection and physical chemistry—might admit simple physical principles or empirical rules. AlphaFold2(Jumper et al., 2021) resolved the practical question definitively, achieving accuracy comparable to experimental methods. The system stores specialized knowledge in approximately 93 million parameters(Wang et al., 2022), trained on roughly 170,000 protein structures—many orders of magnitude smaller than the space of possible sequences. This is the Space-Time

Tradeoff made concrete: learned specialization replacing intractable computation. No simple algorithm or physical principle has subsequently emerged to explain or replicate AlphaFold's success(Skolnick et al., 2021). This precedent suggests that quantum chemistry's structured subset may be similarly tractable only through learned specialization. The analogous training signal—absent for decades—is now available: high-throughput quantum chemistry databases now span $10^5$–$10^6$ configurations across diverse chemical environments (Section 5).

Why ML specifically? Fifty years of human effort produced 400+ functionals without convergence; symbolic regression has not discovered new functional forms. The key is scale: foundation models store specialized knowledge in millions of parameters trained on millions of configurations—a scope neither human intuition nor symbolic search has matched.

**Position 1:** Given uncertainty about the true complexity of chemically relevant problems, the Space-Time Tradeoff implemented via machine learning is the rational path forward. It succeeds across a wide range of scenarios and degrades gracefully when problems are easier than worst-case. Traditional approaches have demonstrably stalled.

### 2.4. The Approximation Workarounds and Their Limits

The field has historically progressed by developing sophisticated approximations to navigate the complexity barrier. Density Functional Theory (DFT) reformulates the problem based on the Hohenberg-Kohn theorems(Hohenberg & Kohn, 1964), which establish that the ground state energy is a unique functional of the electron density, $\rho(\mathbf{r})$:

$$E[\rho] = T[\rho] + V_{ext}[\rho] + V_{ee}[\rho] \tag{2}$$

By working with the three-dimensional density rather than the $3N$-dimensional wavefunction, DFT achieves substantial computational savings. However, the exact form of the exchange-correlation functional $E_{xc}[\rho]$ remains unknown and must be approximated. Wavefunction Theory (WFT) methods take a different approach, systematically approximating the true wavefunction through controlled truncations of the full configuration interaction expansion.

However, these approaches do not bypass intractability; they shift it. Computing the exact $E_{xc}$ in DFT is itself QMA-hard(Schuch & Verstraete, 2009). These methods represent hand-crafted attempts to implement the specialization mandated by the Space-Time Tradeoff, but they are inherently limited by the scope of human design and intuition.

## 3. Quantum Chemistry as Hand-Crafted ML

The historical development of specialized approximations in quantum chemistry was essentially a form of manual, small-scale machine learning that has now reached the limits of human intuition.

### 3.1. Manual Feature Engineering

The selection of mathematical forms and physical ingredients, guided by chemists' and physicists' intuition, is the hallmark of traditional method development. In machine learning terms, this process constitutes manual feature engineering. We argue that this intuition, while historically essential, can act as a limiting bias when it restricts the functional forms explored. However, intuition remains important for defining the scope of the problem and for embedding fundamental physical laws as inductive biases into ML architectures. The challenge lies in discerning when intuition provides essential guidance (such as enforcing symmetries) and when it unnecessarily restricts the hypothesis space.

The development of DFT functionals provides a clear example of this manual feature engineering, well illustrated by the concept of "Jacob's Ladder"(Perdew & Schmidt, 2001). This conceptual hierarchy organizes functionals by their ingredients: the Local Density Approximation (LDA) uses only the local density $\rho(\mathbf{r})$; Generalized Gradient Approximations (GGAs) add the density gradient $\nabla\rho(\mathbf{r})$; meta-GGAs incorporate the kinetic energy density $\tau(\mathbf{r})$; and so forth. Each rung represents chemists acting as feature engineers, manually selecting which physical quantities to include. While successful, this approach is inherently limited by the set of features humans can conceptualize and formulate into analytical equations.

In wavefunction theory, traditional methods involve hand-crafting an approximation (ansatz) for the wavefunction. Coupled Cluster theory, for example, truncates the excitation operator at a chosen level under an exponential ansatz, such as singles and doubles with perturbative triples, yielding CCSD(T)(Raghavachari et al., 1989). The Density Matrix Renormalization Group (DMRG) utilizes tensor network ansatz, pioneered by White (White, 1992) and extensively developed since (Schollwöck, 2011). This manual selection of ansatz structure is directly analogous to choosing model architecture in machine learning.

Furthermore, the representation of the wavefunction almost universally relies on the Linear Combination of Atomic Orbitals (LCAO) approach:

$$\psi_i = \sum_j C_{ij}\phi_j \tag{3}$$

This approach is rooted in the chemical intuition that molecules are composed of atoms(Dunning, 1989). For an ML audience, this strategy of choosing a localized basis is strongly analogous to tokenization in Large Language Models—both are strategies for breaking down a complex, continuous-seeming problem into discrete, meaningful, and computationally efficient units. LCAO exploits the "near-

sightedness" of electronic interactions(Gong et al., 2023; Li et al., 2022), but nothing in the underlying quantum physics mandates this atomic representation. By enforcing this intuition, we introduce a structural bias that may prohibit finding the optimal mathematical representation.

## 3.2. Data-Driven Parameterization

The breakthrough in practical utility often came from empirical parameterization. The iconic DFT functional B3LYP(Becke, 1993) illustrates this clearly: its mixing parameters were optimized using regression against the G1 dataset of approximately 116 data points. In modern terminology, this is supervised learning with a fixed functional form and a small training set.

It is important to acknowledge that traditional methods have seen significant recent progress due to algorithmic advances. DLPNO-CCSD(T)(Riplinger & Neese, 2013) substantially reduces the computational cost of coupled cluster calculations through local correlation approximations. However, these advances improve the efficiency of existing approximations; they do not address the fundamental limitations of the underlying hand-crafted ansatz or functionals.e

## 3.3. Empirical Evidence for Diminishing Returns

The argument that ML represents the most promising path forward gains force from empirical evidence that traditional method development—in both DFT and WFT—has encountered significant barriers.

A landmark study by Medvedev et al.(Medvedev et al., 2017a) analyzed 128 DFT functionals spanning five decades of development from 1974 to 2017. The findings were striking: while energy predictions continued to improve on standard benchmarks, the accuracy of computed electron densities—the fundamental quantity in DFT according to the Hohenberg-Kohn theorems—peaked around 2000 and has since deteriorated. The authors attributed this reversal to "unconstrained functionals sacrificing physical rigor for the flexibility of empirical fitting."

Subsequent debate clarified that while these results apply most directly to atomic systems, they probe a fundamental question: whether functionals are approaching the exact functional or merely achieving good energies through error cancellation (Kepp, 2017; Medvedev et al., 2017b).

Viewed through the lens of our earlier argument, the Medvedev finding takes on deeper significance. Traditional functional development—fitting parameters to minimize errors on benchmark sets—is precisely the "hand-crafted machine learning" we described in Section 2.1. The density regression documented by Medvedev et al. is the signature of this approach breaking down. When a fitting procedure produces better scores on the training metric (energies)

while degrading on the quantity the theory says matters (densities), we recognize a familiar pathology: overfitting, or what economists call Goodhart's Law—optimizing the metric at the expense of the underlying goal. The 400+ functionals represent decades of human-guided "hyperparameter search" that has exhausted the easy gains. The proliferation without convergence suggests saturation: a comprehensive benchmark of 200 functionals (Mardirossian & Head-Gordon, 2017) found that while $\omega$B97M-V emerges as the most balanced choice, "the principal remaining limitations are associated with systems that exhibit significant self-interaction/delocalisation errors and/or strong correlation effects"—precisely the problems ML aims to address.

We do not claim that traditional DFT development has produced no genuine improvements—recent functionals like $\omega$B97M-V and $r^2$SCAN demonstrate real advances on chemically relevant benchmarks. The concern is not absence of progress but its character: each new functional excels in specific domains without convergence toward generality. As Hammes-Schiffer noted in an accompanying perspective(Hammes-Schiffer, 2017), modern functionals "may be giving the correct energies for the wrong reason." Contemporary reviews acknowledge that "DFT methods still do not offer a clear road map for convergence to the exact electronic energy"(Karton & de Oliveira, 2025). This is not a failure of the researchers but a limitation of the paradigm: human intuition operating on benchmark feedback cannot navigate the full functional space systematically.

How is modern neural network-based ML different from this "hand-crafted ML"? The distinction lies in hypothesis space, scale, and constraints. Hand-crafted ML operates within a narrow hypothesis space defined by functional forms humans can conceive—a few dozen parameters encoding intuitions about density gradients and exchange. Modern ML explores hypothesis spaces of millions of parameters, encoding relationships no human could explicitly formulate. Modern ML can also enforce physical constraints as architectural invariants rather than hoping they emerge from fitting. The DM21 functional builds fractional electron constraints into its structure; equivariant networks enforce symmetries exactly. These constraints provide regularization that hand-crafted fitting lacked.

Traditional wavefunction theory faces equally significant barriers. CCSD(T), the celebrated "gold standard," achieves chemical accuracy only for systems "qualitatively well described by a single reference wavefunction"(Bartlett & Musiał, 2007)—a significant caveat that excludes transition metals, bond-breaking processes, excited states, and metallic systems. The perturbative triples treatment becomes inapplicable to three-dimensional metals due to infrared divergence(Liao et al., 2023), and performance on open-shell species is "much more erratic"(Bertels et al., 2021).

The treatment of strongly correlated systems—essential for transition-metal chemistry, catalysis, and materials science—remains, in the words of recent reviews, "still a largely unsolved problem"(Li Manni et al., 2016). The exponential scaling of the Complete Active Space with the number of correlated orbitals creates a hard wall that no amount of algorithmic ingenuity can eliminate.

Recent advances in WFT—DLPNO-CCSD(T), tensor network methods, stochastic approaches—are algorithmic rather than conceptual. They push back the scaling wall but do not remove it. No hand-crafted ansatz has emerged to systematically address strong correlation at scale.

**Position 2:** Machine learning represents the most promising systematic framework for discovering specialized approximations at the scale and generality required for continued progress in quantum chemistry. Traditional method development has encountered significant barriers—with DFT showing signs of benchmark-driven fragmentation and WFT hitting hard walls on both scaling and strong correlation. This is not a proof that ML is logically necessary, but a decision-theoretic argument: given the observed barriers and uncertainty about the problem's true complexity, ML is the rational path forward.

# 4. Interpretability and Chemical Intuition

The argument that human intuition can impose limiting biases on the hypothesis space often raises a concern: the perceived loss of interpretability due to the "black-box" nature of deep neural networks. However, we argue that this represents a necessary trade-off driven by the inherent complexity of the problem.

## 4.1. Reframing Interpretability

It is important to distinguish between different types of understanding. Traditional methods often possess interpretability of construction: a chemist understands the physical concept represented by the gradient term in a GGA functional or the physical meaning of coupled cluster amplitudes. However, many successful traditional methods are not transparent in their parameterization. B3LYP is semi-empirical; the connection between its mixing parameters and the resulting chemical predictions is often opaque, relying on fortuitous error cancellation.

Furthermore, the QMA-hardness of the exact solution suggests that a simple, universally interpretable explanation may not exist. Complexity is an inherent feature of the quantum many-body problem, not an artifact of the ML solution. We must accept that the most accurate models may inherently be the least interpretable by construction.

## 4.2. ML as Instrument for Insight

While we may sacrifice interpretability of construction, ML models still offer two complementary routes to insight. The first is post-hoc interpretation through Explainable AI (XAI): methods like SHAP(Lundberg & Lee, 2017) can identify which atoms or interactions drive a given prediction, though their application to quantum chemistry remains exploratory. The second route—and the deeper claim of this section—is that architecture design itself encodes physical hypotheses, and the model's performance tests them at scale. We develop this view next.

On the second route, a common critique holds that ML has not produced "genuine" physical insight—a new conservation law, a simpler universal functional form, or a previously unknown physical principle. This critique applies an asymmetric standard: traditional computational chemistry has not produced such insights in the past two decades either. The major conceptual frameworks—Hohenberg-Kohn theorems, coupled cluster theory, DMRG—date from decades ago. Recent progress has been algorithmic and parametric rather than conceptual.

Moreover, ML provides a more modest but genuine form of insight: architecture as testable hypothesis. The structure of a neural network encodes physical assumptions, and performance tests those assumptions at scale. E(3)-equivariant architectures dramatically outperform non-equivariant alternatives, confirming that symmetry constraints are essential. Local message passing with 5–6 Å cutoffs succeeds broadly, validating the nearsightedness principle(Prodan & Kohn, 2005). Higher body-order correlations in MACE improve over pair potentials, demonstrating that many-body effects matter. Conversely, when local models fail for ionic systems, this reveals where assumptions break down. The contribution is quantitative rather than conceptual: ML reveals how short-ranged "short-ranged" actually is and which systems violate locality assumptions. Whether ML can move beyond refining known principles to discovering genuinely new ones remains an open frontier. We provide detailed analysis of these quantitative contributions in Appendix C.

The DM21 functional provides a concrete example of how ML can generate insight. Its architecture explicitly enforces correct behavior under fractional electron numbers—a constraint that traditional functionals violate. DM21's success on systems with fractional character, where conventional functionals fail severely, validates that these mathematical constraints correspond to genuine physical requirements. The architecture choice led to physical insight: fractional electron behavior is not an edge case but a fundamental test of functional quality.

**Position 3:** ML provides physical insight through architecture validation: successful designs confirm physical princi-

ples (symmetry, locality, many-body effects) while failures reveal their limits. Combined with evolving XAI techniques, ML offers a pathway to transform models from passive predictors into active hypothesis generators.

# 5. The Catalysts for the ML Revolution

If machine learning represents the most promising path for progress in quantum chemistry, why is this revolution happening now rather than decades ago? The answer lies in the convergence of three technological developments that have made the automation of approximation both feasible and necessary.

The first catalyst is the explosion of high-throughput quantum chemistry data. Landmark databases like QM9(Ramakrishnan et al., 2014) (134,000 organic molecules), the Materials Project(Jain et al., 2013) (hundreds of thousands of inorganic materials), and MPtrj(Batatia et al., 2025) (1.6 million configurations spanning 89 elements) provide the training signal for ML models to discover patterns in quantum mechanical behavior.

The second catalyst is the development of neural network architectures with appropriate inductive biases for chemistry. Generic neural networks struggle with molecular data because they do not respect fundamental physical symmetries. Graph Neural Networks(Gilmer et al., 2017) represented a crucial advance by operating directly on molecular graphs, treating atoms as nodes and bonds as edges. The subsequent integration of physical symmetries—particularly rotational and translational equivariance(Batzner et al., 2022)—into architectures like NequIP, MACE, PaiNN, Allegro(Musaelian et al., 2023), and eSCN(Passaro & Zitnick, 2023) made models dramatically more data-efficient. By building in the constraint that predictions should not change when a molecule is rotated or translated, these architectures can learn from smaller datasets and generalize more reliably. This represents a productive collaboration between human intuition (identifying the relevant symmetries) and machine learning (learning everything else).

# 6. Machine Learning as Automated Specialization

Given the promise of specialized approximations (Position 1) and the technological catalysts described above, machine learning provides the framework to systematically navigate the complex landscape of approximations. ML automates the discovery of specialized strategies and encodes them into model parameters (weights), effectively implementing the Space-Time Tradeoff at scale.

**Position 4:** Machine learning provides the tools to automate the "hand-crafting" of the past, enabling the systematic discovery, storage, and dynamic adaptation of specialized solutions by exploring functional and architectural spaces far beyond the limits of manual design.

## 6.1. Dynamic Specialization

Traditional methods apply approximations uniformly. Machine learning enables dynamic specialization: in active learning molecular dynamics(Li et al., 2015; Kulichenko et al., 2023), models estimate uncertainty and automatically request new reference data when entering novel configurations. This real-time adaptation is unique to ML and beyond the capabilities of fixed approximations.

## 6.2. Hybrid Approaches

The transition to data-driven models is not binary. Machine Learned Potentials (MLPs) learn potential energy surfaces from quantum chemical data, achieving near-quantum accuracy at costs orders of magnitude lower(Noé et al., 2020). Architectures like MACE(Batatia et al., 2022) simulate millions of atoms over microsecond timescales. This hybrid strategy extends beyond ground-state energies: Ren et al.(Ren et al., 2025) combined the Frenkel exciton model with ML-predicted Hamiltonian elements to predict optical properties of 50-monomer aggregates—demonstrating how ML extends established physical models.

## 6.3. Foundation Models

A major trend enabled by large datasets and expressive architectures is the rise of foundation models in chemistry. These models are trained on diverse data spanning vast regions of chemical space, aiming to capture general quantum mechanical principles rather than system-specific behavior.

The MACE-MP-0 model(Batatia et al., 2025) exemplifies this approach for interatomic potentials. Trained on 1.6 million configurations from the Materials Project spanning 89 elements, this foundation model achieves useful accuracy (approximately 20 meV/atom for energies) across an enormous range of chemical systems without any system-specific training. Similarly, the PET-MAD-DOS model(How et al., 2025) demonstrates this approach for electronic properties, predicting the Density of States for systems ranging from small molecules to complex materials.

These foundation models enable highly efficient transfer learning. Fine-tuning on a small amount of system-specific data typically achieves accuracy comparable to or exceeding bespoke models trained from scratch on much larger datasets. This pre-train-then-fine-tune paradigm offers a scalable and data-efficient pathway to generate the specialized approximations we have argued are necessary.

This foundation model paradigm is now extending beyond

interatomic potentials to neural network wavefunctions themselves. Orbformer(Foster et al., 2025), pretrained on diverse molecular structures including bond-breaking configurations, demonstrates that the pre-train-then-fine-tune approach can achieve high accuracy for multireferential systems—precisely the regime where traditional wavefunction methods face their steepest computational barriers.

The practical implications are substantial. ML potentials achieve DFT-competitive accuracy at computational costs $10^4$–$10^6$ times lower—milliseconds versus minutes or hours per evaluation. This speedup enables qualitatively new applications: screening millions of candidate molecules for drug discovery, simulating protein dynamics over microseconds, exploring vast configuration spaces for materials design. These applications were simply intractable with traditional quantum chemistry, regardless of algorithmic improvements. The economic case for ML is not merely that it matches traditional accuracy more efficiently, but that it unlocks capabilities that no amount of DFT optimization could provide.

### 6.4. Overcoming the Medvedev Paradox

Machine learning excels where manual design becomes intractable: integrating complex constraints with flexible data fitting. The Medvedev paradox arose because traditional functional development—hand-crafted ML optimizing benchmark energies—lacked the regularization that physical constraints provide. Without such constraints, the optimization found spurious solutions that improved energies while degrading densities.

DeepMind's DM21 functional(Kirkpatrick et al., 2021) demonstrated how principled ML overcomes this failure mode. The DM21 paper explicitly cites the Medvedev et al. study, acknowledging the crisis in traditional functional development as motivation. DM21 was trained to obey exact mathematical conditions—particularly correct behavior under fractional electron numbers and fractional spin—that traditional functionals routinely violate. These constraints act as regularization, preventing the overfitting that plagued benchmark-driven development. By learning from data while satisfying physical constraints, DM21 achieved what decades of hand-crafted fitting could not: correctly describing systems with fractional electron character. The contrast is instructive: unconstrained fitting on benchmarks led to the Medvedev paradox; constrained learning from data provides a path beyond it.

We note, however, that DM21 has not yet seen widespread adoption despite its benchmark performance. The barriers are instructive: SCF convergence failures for transition metals, lack of analytic gradients for geometry optimization, computational overhead that can exceed CCSD(T) for small molecules, and availability only in PySCF rather than widely-used packages like Gaussian or VASP. These are

engineering challenges, not fundamental limitations—the kind that sustained investment could overcome. B3LYP's dominance reflects decades of software integration and accumulated trust, not inherent superiority. We provide detailed analysis of this adoption gap in Appendix B.

### 6.5. The Wavefunction Frontier

Machine learning enables exploration of ansatz structures far beyond human design. FermiNet(Hermann et al., 2020) represents a neural network wavefunction that enforces fermionic antisymmetry while operating in continuous real space, completely free from atom-centered basis sets. By bypassing the LCAO framework, it explores a vastly wider hypothesis space, achieving high accuracy on small molecules and atoms.

Recent work has begun to address the transferability limitation that initially constrained these methods to per-system training. Scherbela et al.(Scherbela et al., 2024) demonstrated that neural network orbitals can transfer across multiple compounds and geometries by mapping Hartree-Fock orbitals to correlated neural network orbitals. This approach has been extended to solids, where Cassella et al.(Gerard et al., 2025) achieved 50-fold reductions in optimization cost by transferring networks trained on small supercells to larger ones. Most ambitiously, the Orbformer foundation model(Foster et al., 2025)—pretrained on 22,000 equilibrium and dissociating structures—demonstrates that transferable neural wavefunctions can accurately describe chemical bond breaking, a difficult multireferential problem. These developments suggest that neural network wavefunctions are following the same trajectory as neural network potentials: from per-system training toward foundation models that generalize across chemical space.

Neural scaling laws for wavefunction methods(Jiang et al., 2025) show that accuracy improves predictably with model size. This suggests the path to higher accuracy is clear: invest in larger models rather than hoping for conceptual breakthroughs in ansatz design.

## 7. The Quantum Computing Frontier

Will fault-tolerant quantum computers render AI approaches obsolete? We argue no—AI will become an indispensable partner to quantum computing.

The QMA-hardness results apply to quantum computers as well: even ideal quantum hardware faces fundamental limitations on arbitrary systems. Near-term algorithms like VQE face practical challenges including barren plateaus and noise-induced errors. More fundamentally, quantum algorithms solve the electronic structure problem for single nuclear geometries, while real applications require energies at millions of configurations for dynamics and thermody-

namics. Using quantum resources for every configuration would be highly inefficient.

This suggests a natural division of labor: quantum computers will serve as the ultimate reference method for difficult systems where classical methods fail; ML will generalize from limited quantum calculations to the vast configuration space needed for practical applications.

**Position 5:** Quantum computers will generate high-quality reference data for difficult systems. AI will be the essential generalization engine that makes this data practically useful for dynamics, thermodynamics, and materials discovery.

## 8. The ML-Driven Paradigm Shift in Action

The theoretical arguments for the promise of ML (Positions 1–4) are supported by the current trajectory of the field. ML is not merely improving existing methods but unlocking capabilities that were intractable with traditional approaches.

### 8.1. Orbital-Free DFT

Kohn-Sham DFT reintroduces orbitals to compute kinetic energy, increasing cost to $O(N^3)$ and limiting applications to thousands of atoms. Orbital-Free DFT promises linear scaling by eliminating orbitals entirely, but decades of effort failed to hand-craft an accurate kinetic energy functional.

Two recent ML-based approaches surmounted this challenge. M-OFDFT (Zhang et al., 2024) uses a deep learning kinetic energy functional to achieve Kohn-Sham accuracy on a wide range of molecules and extrapolates to systems much larger than those seen during training. STRUCTURES25 (Remme et al., 2025) uses equivariant neural networks trained on diverse configurations to obtain a stably convergent functional with chemical accuracy. Both validate ML's power to discover relationships that eluded decades of intuition-driven design.

### 8.2. Excited States via Deep VMC

Excited states—essential for photochemistry and spectroscopy—are particularly challenging for traditional WFT, which scales prohibitively. Deep Variational Monte Carlo leverages neural network wavefunctions to achieve WFT-level accuracy with $O(N^{3-4})$ scaling versus $O(N^7)$ for CCSD(T). Entwistle et al.(Entwistle et al., 2023) extended PauliNet to excited states, successfully modeling conical intersections that are difficult for standard methods.

## 9. Challenges and Limitations

Intellectual honesty requires acknowledging significant challenges that ML in quantum chemistry must overcome. These include: the accuracy-generality trade-off (universal models sacrifice precision for breadth); long-range interactions beyond local cutoffs; reactivity and rare events underrepresented in training data; out-of-distribution generalization and the risk of silent failures; wavefunction method transferability (though recent foundation models like Orbformer(Foster et al., 2025) show promising progress toward transfer learning); and the gap between benchmark success and production adoption exemplified by DM21. Appendix B provides detailed discussion of each challenge.

These define active research frontiers with clear paths forward—larger models, better architectures, improved uncertainty quantification, software engineering—rather than fundamental barriers. The alternative, continued reliance on hand-crafted methods that have demonstrably stalled, offers no comparable roadmap for progress.

## 10. Alternative Views

We have argued that machine learning represents the most promising path forward for quantum chemistry. Here we consolidate and directly engage with the strongest counterarguments to this position.

### 10.1. The Typical-Case Objection

**Alternative view:** QMA-hardness is a worst-case result. Real chemistry may be tractable without ML.

**Response:** We acknowledge this remains formally unproven. However, fifty years of sustained effort by the quantum chemistry community provides strong empirical evidence that the chemically relevant subset does not admit simple analytical solutions: 400+ DFT functionals have been developed without convergence toward universality; density accuracy has deteriorated since 2000 (Medvedev et al., 2017a); strong correlation remains unsolved despite extensive work; and no tractable analytical breakthrough has emerged for broad chemical space. This track record does not prove the relevant subset is QMA-hard, but it does suggest the subset resists hand-crafted specialization—which is what makes the decision-theoretic case for ML (Section 1.3) compelling rather than merely abstract. The AlphaFold precedent—where NP-hard protein folding was solved through learned specialization, not analytical insight—supports this conclusion.

### 10.2. The Interpretability and Insight Objection

**Alternative view:** ML has not produced genuine physical insight—no new conservation laws or universal functional forms.

**Response:** This applies an asymmetric standard: traditional methods have not produced conceptual advances in recent decades either. As discussed in Section 3.2,

ML provides insight through architecture validation—E(3)-equivariance confirms symmetry's importance; 5–6 Å cut-offs quantify nearsightedness; body-order scaling reveals correlation structure. The contribution is quantitative rather than conceptual, but no less genuine.

### 10.3. The Quantum Computing Objection

**Alternative view:** Fault-tolerant quantum computers will render AI approaches obsolete.

**Response:** QMA-hardness applies to quantum computers as well. More fundamentally, as argued in Section 6, quantum algorithms solve single geometries while applications require millions of configurations. Position 5 articulates the natural division of labor: quantum computers as reference generators, ML as the generalization engine.

### 10.4. The Continued Progress Objection

**Alternative view:** Traditional methods continue to advance—new functionals show improvements, tensor network methods push tractable sizes. Reports of stagnation are premature.

**Response:** We do not deny improvement, but question whether it addresses fundamental barriers. DLPNO-CCSD(T) reduces prefactors without changing $O(N^7)$ scaling; new functionals show fragmentation without convergence toward universality. Recent advances are algorithmic rather than conceptual. ML offers systematic exploration of larger hypothesis spaces with principled physical constraints.

## 11. Implications for Research Priority

Four concrete shifts in research priority follow from the position above. First, funding agencies should prioritize ML infrastructure—datasets, equivariant architectures, uncertainty quantification, and benchmarking—over incremental traditional method development. Second, quantum chemistry software packages should treat ML integration as a first-class capability rather than an afterthought: the DM21 adoption gap (Appendix B.6) demonstrates that benchmark success without software integration produces little real-world progress. Third, the community should establish standard benchmark protocols that go beyond average accuracy to include out-of-distribution generalization and per-prediction reliability reporting—essential for trustworthy deployment given the silent-failure mode discussed in Appendix B.4. Fourth, graduate training in computational chemistry should emphasize ML methodology alongside traditional approaches, preparing researchers for the paradigm shift underway.

## 12. Conclusion

Machine learning at scale offers a path beyond possible saturation. This is not a claim of logical necessity but a decision-theoretic argument: ML succeeds across the range of plausible scenarios, while traditional approaches have demonstrably stalled. What would falsify this position? A non-ML approach achieving chemical accuracy across broad chemical space, with favorable scaling and systematic treatment of strong correlation—without extensive parameterization against reference data. We consider this unlikely given fifty years of effort, but acknowledge it as a logical possibility.

## Acknowledgments

We acknowledge support from Academia Sinica (Grant No. AS-IV-114-M01), National Science and Technology Council of Taiwan (Grant Nos. NSTC 1114-2113-M-001-018 and 114-2113-M-001-022).

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

# A. Quantitative Analysis of the Space-Time Tradeoff

The Space-Time Tradeoff (TSTO) principle, which forms a central pillar of our argument, can be grounded in concrete quantitative terms. This appendix provides a detailed analysis of model sizes, accuracies, and the compression ratios achieved by modern machine learning approaches to quantum chemistry.

## A.1. The Scale of Chemical Space

Before examining what ML models achieve, it is instructive to consider the scale of the problem they address. The size of "drug-like" chemical space—the ensemble of small organic molecules that might serve as pharmaceuticals—has been the subject of considerable study. Early estimates (Bohacek et al., 1996) placed this number at approximately $10^{60}$ molecules when considering all possible structures obeying Lipinski's rules for oral bioavailability with molecular weight below 500 Daltons. More recent work by (Reymond, 2015), based on systematic enumeration of molecules up to 17 heavy atoms in the GDB-17 database, suggests that realistic estimates accounting for synthetic feasibility and chemical stability fall in the range of $10^{33}$ molecules.

Even the more conservative estimate represents a number far beyond any possibility of exhaustive enumeration or tabulation. For comparison, the number of atoms in the observable universe is approximately $10^{80}$, and the number of possible chess games is estimated at $10^{120}$. The chemical space relevant to materials science, including inorganic compounds, alloys, and extended systems, is larger still.

The quantum mechanical description of each molecule in this space requires solving the electronic Schrödinger equation, a problem whose computational cost scales exponentially with system size for exact methods. Traditional approaches to this challenge have relied on developing approximate methods—DFT functionals, coupled cluster truncations, and the like—that trade accuracy for tractability. The question is whether there exists a more systematic approach to navigating this vast space.

## A.2. Model Sizes and Achieved Accuracies

Modern machine learning potentials and functionals achieve high accuracy with modest numbers of parameters. The following examples illustrate the state of the art as of 2025.

The MACE-MP-0 foundation model (Batatia et al., 2025), trained on 1.6 million configurations from the Materials Project database, contains approximately 4.69 million parameters in its medium variant. This model achieves mean absolute errors of roughly 20 meV per atom for energies and 45 meV per Ångstrom for forces across 89 chemical elements. While this accuracy falls short of the strict "chemical accuracy" threshold of 1 kcal/mol (approximately 43 meV), it is sufficient for many applications including structure optimization, phase stability prediction, and molecular dynamics simulation. The larger MACE-MPA-0 model, with 9.06 million parameters and trained on an expanded dataset including the Alexandria database, achieves state-of-the-art performance on materials discovery benchmarks.

For organic molecules specifically, the MACE-OFF23 models demonstrate that chemical accuracy is achievable with current methods. The large variant achieves errors of approximately 0.5–1.0 meV per atom, well below the 43 meV threshold, for drug-like organic molecules. These models are sufficiently accurate to reproduce experimental thermodynamic properties including densities, enthalpies of vaporization, and conformational preferences.

The AlphaFold2 protein structure prediction system provides a useful point of comparison from computational biology. This model contains approximately 93 million parameters and was trained on roughly 170,000 experimentally determined protein structures (Jumper et al., 2021). It achieves accuracy comparable to experimental methods on most protein targets, as demonstrated in the CASP14 competition. The contrast between the training set size (approximately $10^5$ structures) and the space of possible protein sequences illustrates the compression that learned representations can achieve when the underlying data has exploitable structure.

### A.3. Compression Ratios and What They Imply

The numbers above imply compression ratios that merit consideration. If we take the conservative estimate of $10^{33}$ for drug-like chemical space and note that models achieving chemical accuracy contain on the order of $10^7$ parameters, we obtain a compression ratio of approximately $10^{26}$. Using the more expansive estimate of $10^{60}$ yields a compression ratio of $10^{53}$.

These compression ratios are only possible because chemistry possesses exploitable structure that learned representations can capture. The key structural features include locality (the energy contribution of an atom depends primarily on its local environment), smoothness (similar configurations have similar energies), and symmetry (energies are invariant to rotations, translations, and permutations of identical atoms). Neural network architectures that build in these properties—particularly equivariant graph neural networks—

can leverage this structure to achieve generalization far beyond what would be expected from naive interpolation.

It is important to note that these models do not memorize chemical space. A lookup table for $10^{33}$ molecules, even storing only a single energy value per molecule at 64-bit precision, would require approximately $10^{34}$ bytes of storage—more than the information content of all human civilization. Instead, the models learn generalizable patterns that enable accurate interpolation and, to a lesser extent, extrapolation across chemical space.

### A.4. Neural Scaling Laws in Chemistry

Recent work has begun to characterize how model accuracy scales with model size and training data in chemistry applications, analogous to the scaling laws extensively studied for large language models.

(Frey et al., 2023) investigated neural scaling behavior for large chemical models, varying model and dataset sizes over many orders of magnitude. For chemistry language models, they observed scaling exponents of approximately 0.17 for the largest dataset sizes considered, indicating that doubling model size yields approximately 12% improvement in performance metrics. For graph neural networks used as interatomic potentials, the scaling behavior depends on both model architecture and the degree to which physical symmetries are incorporated.

More recent work has demonstrated neural scaling laws specifically for wavefunction methods. (Jiang et al., 2025) showed that absolute energy errors in neural network quantum Monte Carlo calculations decrease as a power law with increasing model capacity, following the relationship $\epsilon \propto N^{-\alpha}$ where $N$ is the number of parameters and $\alpha$ is a system-dependent exponent. This scaling behavior enabled them to achieve sub-chemical-accuracy results on challenging systems including transition states and strongly correlated molecules.

These scaling laws have important practical implications. They suggest that the path to higher accuracy is clear and predictable: invest in larger models and more training data. This stands in contrast to traditional method development, where progress has been sporadic and unpredictable, often requiring conceptual breakthroughs that cannot be scheduled or guaranteed.

### A.5. Caveats and Limitations

Several important caveats apply to the quantitative analysis presented above.

First, the accuracy figures quoted for universal potentials like MACE-MP-0 represent averages across diverse test sets. Performance on specific systems, particularly those far

from the training distribution, may be significantly worse. Fine-tuning on system-specific data typically improves accuracy substantially but requires additional computational investment. For applications requiring guaranteed accuracy bounds, careful validation on relevant test cases is essential.

Second, the chemical space size estimates span many orders of magnitude depending on the assumptions made. The $10^{60}$ estimate includes many molecules that are synthetically inaccessible or chemically unstable, while the $10^{33}$ estimate applies specifically to drug-like molecules and excludes larger systems, inorganic materials, and extended structures. The appropriate comparison depends on the intended application domain.

Third, achieving chemical accuracy for energies does not guarantee accuracy for all properties of interest. Forces, which are derivatives of energies, are typically predicted less accurately. Properties involving electronic excited states, spin states, or response to external fields may require specialized models or additional training data. The reliability of uncertainty estimates produced by these models is an active area of research.

Fourth, the computational cost of training and inference, while much lower than *ab initio* methods, is not negligible. Training a foundation model from scratch requires substantial GPU resources, and inference with large neural networks is slower than evaluation of classical force fields. These costs must be weighed against the accuracy benefits for specific applications.

Despite these caveats, the quantitative evidence supports the central thesis: machine learning enables a practical implementation of the Space-Time Tradeoff that achieves useful accuracy across vast regions of chemical space with tractable computational and storage requirements.

# B. Detailed Discussion of Challenges

This appendix provides extended discussion of the challenges outlined in Section 8, offering context for practitioners and researchers working to address these limitations.

### B.1. The Accuracy-Generality Trade-off

Appendix A details the accuracy achieved by current models. The key tension is that universal models sacrifice accuracy for generality, while specialized models sacrifice generality for accuracy. Practitioners must navigate this trade-off based on their specific applications, often employing transfer learning to fine-tune universal models on domain-specific data.

### B.2. Long-Range Interactions

Most ML potential architectures use local cutoffs of 5–6 Ångstroms, which fail to capture long-range Coulomb interactions in ionic systems and polar solvents. Appendix C discusses how this limitation itself provides physical insight about where locality assumptions break down. From a practical standpoint, several remedies are under development: Ewald summation for periodic systems, learned charge equilibration schemes, and hierarchical architectures. However, no approach has yet achieved the combination of accuracy, efficiency, and generality required for routine use across diverse chemical systems.

### B.3. Reactivity and Rare Events

Chemical reactivity poses challenges that go beyond accuracy on equilibrium structures. Transition states, which determine reaction rates and mechanisms, occupy saddle points on the potential energy surface. Standard training datasets, generated by sampling around equilibrium configurations, systematically underrepresent these critical regions. Bond breaking and formation involve qualitative changes in electronic structure—from covalent to ionic character, changes in spin state, or electron transfer—that models trained on equilibrium data may not capture.

These limitations are particularly problematic for applications in catalysis, where transition state energies determine selectivity, and for reaction mechanism discovery, where the goal is precisely to explore regions of configuration space far from equilibrium. Active learning approaches that specifically target transition states and reactive regions show promise but require careful design of acquisition functions and significant computational investment in reference calculations.

### B.4. Out-of-Distribution Generalization

Perhaps the most fundamental challenge is reliable behavior on systems outside the training distribution. Machine learning models are fundamentally interpolators; their predictions are reliable within the manifold of configurations similar to training data but can fail on novel systems. The combinatorial vastness of chemical space means any finite training set covers only a tiny fraction of possible configurations.

Unlike traditional *ab initio* methods, which are grounded in physical principles that apply universally (even if expensive to evaluate), ML models can fail silently—producing confident but incorrect predictions on out-of-distribution inputs. For high-stakes applications, such failures could lead to wasted experimental resources pursuing false positives or, worse, safety issues from undetected errors.

Reliable uncertainty quantification—the ability to know when the model doesn't know—is essential but remains an active research challenge. Ensemble methods, Monte Carlo dropout, and evidential deep learning provide uncer-

tainty estimates, but calibrating these estimates to be reliable across diverse chemical systems is difficult. The field increasingly recognizes that uncertainty quantification is not optional but essential for trustworthy deployment.

### B.5. Wavefunction Method Transferability

Neural network wavefunction methods like FermiNet and PauliNet achieve high accuracy by learning flexible representations of the many-electron wavefunction. A significant limitation has been that, unlike neural network potentials that transfer across chemical space after training, these methods historically required training from scratch for each new molecular system—making the computational cost per molecule comparable to or exceeding traditional coupled cluster calculations.

However, recent breakthroughs have begun to address this limitation systematically. Scherbela et al.(Scherbela et al., 2024) proposed a neural network ansatz that maps uncorrelated Hartree-Fock orbitals to correlated neural network orbitals, enabling a single wavefunction model to transfer across multiple compounds and geometries. Building on this approach, Cassella et al.(Gerard et al., 2025) extended transferable neural wavefunctions to solids, demonstrating that pretraining on small supercells ($2{\times}2{\times}2$ LiH) and fine-tuning on larger systems ($3{\times}3{\times}3$) reduces the required optimization steps by a factor of 50 compared with training from scratch.

Most significantly, Foster et al.(Foster et al., 2025) introduced Orbformer, an ab initio foundation model of wavefunctions pretrained on 22,000 equilibrium and dissociating molecular structures. Orbformer can be fine-tuned on unseen molecules to achieve accuracy rivalling classical multireferential methods at a fraction of the computational cost. This model accurately describes chemical bond breaking—a regime where the multireferential character of electronic structure has historically posed severe challenges for both traditional methods and earlier neural network approaches.

These developments suggest that neural network wavefunction methods are following the trajectory established by neural network potentials: progressing from per-system optimization toward foundation models that amortize training cost across chemical space. The combination of FermiNet-level accuracy with MACE-level transferability—previously identified as an "important open challenge"—is becoming increasingly achievable.

### B.6. The Adoption Gap: A DM21 Case Study

Benchmark success has not yet translated to widespread production use for many ML methods. DM21 provides an instructive case study: despite benchmark-leading performance and explicit addressing of the fractional electron

problem documented by Medvedev et al., it sees limited use in routine computational chemistry. The reasons illuminate broader adoption barriers.

SCF convergence failures: DM21 struggles to achieve self-consistent field convergence for transition metal complexes—the systems where improved functionals are most needed. Studies report that SCF convergence represents "the major obstacle to the use of DM21 in TMC applications," requiring elaborate multi-strategy convergence protocols that standard functionals do not need.

Computational overhead: Counter-intuitively, DM21 can be slower than CCSD(T) for small molecules due to the need to compute two-electron integrals at each grid point and the difficulty of SCF convergence. The promised efficiency gains materialize only for larger systems where the $O(N^3)$ scaling advantage dominates.

No analytic gradients: Geometry optimization—the most common DFT task—requires energy gradients. DM21's neural network produces oscillatory gradients that cause optimization failures, particularly for configurations far from equilibrium. This limitation excludes most practical workflows.

Software availability: DM21 is available only in PySCF via a non-standard interface, not in widely-used packages like Gaussian, ORCA, or VASP. Integration with standard features like dispersion corrections requires workarounds. No periodic boundary condition support exists.

Generalization concerns: Independent analysis questioned whether DM21's benchmark success reflects genuine understanding of fractional electron systems or memorization of training data, with one study concluding that DM21 "can easily get away with memorizing" on key test sets.

These barriers are engineering challenges, not fundamental limitations—the kind that sustained investment could overcome. They illustrate why this paper argues for systematic development rather than one-off demonstrations. B3LYP's dominance reflects decades of software integration, validated workflows, and accumulated trust, not inherent superiority. ML methods must build comparable infrastructure.

### B.7. Data Quality and Provenance

ML models inherit the limitations of their training data. When trained on DFT calculations, models cannot systematically exceed DFT accuracy; they learn to interpolate within the reference method's error distribution. This is a fundamental constraint distinct from the generalization challenges discussed above.

However, this limitation underscores that ML and traditional methods are complementary rather than competitive. High-quality traditional calculations—CCSD(T), DMRG, quan-

tum Monte Carlo, and more recently large-scale $r^2$SCAN datasets—remain essential for generating accurate reference data. Recent work demonstrates this paradigm concretely: MP-ALOE (Kuner et al., 2025) and the foundational PES dataset of Kaplan et al. (Kaplan et al., 2025) provide $r^2$SCAN training data for universal ML potentials, and Kim et al. (Kim et al., 2026) show that the resulting potentials transfer effectively across chemical domains. The growing availability of such data (ANI datasets, SPICE, W4-17) enables ML models trained on increasingly reliable references. Multi-fidelity approaches (Fernández-Godino, 2023) that combine abundant low-accuracy data with sparse high-accuracy calculations offer a principled path forward. The argument for ML investment does not imply abandoning traditional methods; rather, it reframes their role from direct application toward data generation for ML training.

## C. Quantitative Validation of Architectural Insights

Section 3.2 argues that successful neural network architectures constitute physical insight by validating and refining known principles. This appendix provides detailed analysis of these quantitative contributions.

### C.1. Locality and Cutoff Distances

The nearsightedness principle of electronic matter, established by (Prodan & Kohn, 2005), holds that local properties depend primarily on the nearby environment. ML architectures quantify this principle. Most successful interatomic potentials use message-passing schemes with cutoffs of 5–6 Ångstroms, and the accuracy achieved with such short cutoffs (see Appendix A for specific numbers) confirms that electronic interactions are short-ranged for most applications.

Equally informative are the failures. Local architectures systematically underperform on ionic systems, charged molecules, and polar solvents where long-range Coulomb interactions dominate. The accuracy degradation in these systems is not subtle: errors can increase by factors of 2–5 compared to neutral systems. This quantifies where the locality assumption breaks down and motivates hybrid approaches combining local neural networks with explicit long-range electrostatics.

### C.2. Many-Body Correlations and Body Order

Traditional interatomic potentials often truncate at the pair level, with three-body corrections added empirically (e.g., Axilrod-Teller terms). ML architectures systematically test which correlation orders matter. The MACE architecture (Batatia et al., 2022) explicitly constructs many-body messages up to specified body order, enabling controlled experi-

ments. Empirically, four-body correlations provide substantial accuracy improvements over three-body models for bulk materials and surfaces. For molecular systems, three-body terms typically suffice for equilibrium properties, but four-body contributions become essential near transition states and for accurate forces.

This hierarchy was not obvious from physical intuition alone. While chemists understood that many-body effects exist, the quantitative importance of four-body versus three-body versus pair interactions for different properties and systems was established empirically through architectural comparisons. The finding that accuracy scales predictably with body order—rather than saturating at low order—validates the physical importance of higher correlations and guides architecture design.

### C.3. Symmetry and Data Efficiency

The most significant architectural insight concerns equivariance. Non-equivariant neural networks can, in principle, learn rotational invariance from data augmentation. In practice, E(3)-equivariant architectures achieve equivalent accuracy with 10–100$\times$ less training data. This efficiency gain quantifies how fundamental symmetry is to the problem: the constraint eliminates vast regions of function space that would otherwise require data to rule out.

Architectures also reveal which symmetries matter most. Permutation equivariance (over identical atoms) is essential; models lacking it fail entirely. Rotational equivariance provides the largest efficiency gains. Translation invariance, typically enforced by using relative positions, is necessary but less informative since it was already universally assumed. The hierarchy of importance—permutation > rotation > translation—was established empirically.

### C.4. Comparison of Methods

Table 1 synthesizes quantitative comparisons between traditional and ML methods, drawing on the model details in Appendix A. The key observations: ML potentials achieve accuracy competitive with DFT at costs four to six orders of magnitude lower; foundation models cover 89 elements with a single model; and neural network wavefunction methods match CCSD(T) accuracy with better scaling. These comparisons support the thesis that ML provides a practical implementation of the Space-Time Tradeoff.

*Table 1.* Comparison of traditional and machine learning methods for quantum chemistry.

| Method | Accuracy | Scaling | Generality | Cost |
|---|---|---|---|---|
| CCSD(T) | $\sim$1 kcal/mol | $O(N^7)$ | Single-reference | Hours |
| DFT (B3LYP) | 3–5 kcal/mol | $O(N^3)$ | Broad | Minutes |
| DFT ($\omega$B97M-V) | 2–3 kcal/mol | $O(N^3)$ | Thermochemistry | Minutes |
| MACE-MP-0 | $\sim$20 meV/atom | $O(N)$ | 89 elements | Milliseconds |
| MACE-OFF23 | <1 meV/atom | $O(N)$ | Organic | Milliseconds |
| ANI-2x | $\sim$3 kcal/mol | $O(N)$ | CHNO | Milliseconds |
| FermiNet | $\sim$1 kcal/mol | $O(N^{3-4})$ | Per-system | Hours |

