# OpenReview forum: "Position: The Inevitable Transition to Machine Learning in Quantum Chemistry"
_ICML.cc/2026/Position_Paper_Track — ICML 2026 Position Paper Track regular_

### Official Review · Reviewer_eCaa · 2026-02-26

**Significance:** 3
**Argument Clarity:** 4
**Rating:** 5
**Confidence:** 5

**Questions:**

Scaling up data and model size seems like a promising route for advancing quantum chemistry, but high-level QC methods (e.g., FCI) are typically infeasible for large molecules. This raises two concerns: (1) ML models may not have access to enough high-fidelity labels to exhibit meaningful neural scaling laws, and (2) even with substantial data, model accuracy may be fundamentally capped by the fidelity of the labeling method used to generate the training set. How do you address these two inherent limitations of ML-based approaches?

**Alternative Views Section:**

Yes

**Compliance With Llm Reviewing Policy A Conservative:**

Affirmed.

**Discussion Potential:**

3

**Paper Summary:**

This Position Paper proposes a decision-theoretic argument that machine learning (ML) is the most promising path for progress in quantum chemistry.
It frames the central difficulty of quantum chemistry (QC) through computational complexity and a space–time tradeoff, arguing that universally applicable and interpretable analytic methods are unlikely to overcome QMA-hard worst-case instances in QC. This motivates “stored specialization” in learned model parameters as a more rational strategy under uncertainty. To support this view, the authors reintroduce decades of DFT functional development and wavefunction ansatz design as “hand-crafted machine learning,” suggesting that marginal returns are saturating—particularly for density fidelity and strongly correlated regimes. As contributions, the paper connects QC approximation to space–time tradeoff strategies in computer science, and presents a set of explicit position statements spanning DFT, ML interatomic potentials, neural-network wavefunctions, and foundation-model pretraining. It further argues that modern ML can scale the search over hypothesis spaces while incorporating physical constraints to improve robustness and reduce benchmark-driven pathologies. Finally, it outlines calls to action—e.g., scaling data and models, strengthening constraints and uncertainty estimation, and improving software integration, to move ML approaches from benchmark success to reliable deployment.

**Position:**

Yes

**Position In Title:**

Yes

**Related Work:**

3

**Strengths And Weaknesses:**

* Strengths
  * The paper states a clear and consistent position, and the use of structured “position statements” makes the thesis easy to follow.
  * It offers a coherent, unifying framework—addressing complexity limits via a space–time tradeoff realized as learned specialization—and grounds the discussion in both the historical trajectory of DFT development and recent AI+QC advances. This makes the topic timely, relevant to the ICML community, and likely to spark constructive debate.
  * The presentation is generally well organized and clearly written, and it engages alternative viewpoints rather than dismissing them, which strengthens its potential to inspire discussion even among readers who disagree.

* Weaknesses
  * The strongest claim—an “inevitable transition”—is not as tightly supported as the more defensible claim that ML is a rational, high-priority strategy. In particular, the step from worst-case complexity results to “chemistry-relevant” instances remains suggestive rather than conclusive.

  * Much of the support is trend- and case-study–based; the paper would benefit from more quantitative synthesis that would allow readers to evaluate the position against future outcomes.

  * The call to action remains somewhat high-level (e.g., scale data/models, add constraints, improve tooling). It would be stronger with more concrete, ICML-style milestones—such as specific benchmark suites, evaluation protocols (including reliability and OOD/generalization reporting), and clearer discussion of governance and failure modes.

  * In the discussion of ML-based orbital-free DFT, an important recent reference appears to be missing: M-OFDFT[1], which demonstrates the advantages of ML for improving OFDFT accuracy on 3D molecular systems.

[1] Zhang H, Liu S, You J, et al. Overcoming the barrier of orbital-free density functional theory for molecular systems using deep learning[J]. Nature Computational Science, 2024, 4(3): 210-223.

**Support:**

2

---

> ### Author Rebuttal · Authors · 2026-03-29
>
> We thank the reviewer for these precise and constructive criticisms.
>
> We acknowledge that "inevitable" in the title is deliberately stronger than the decision-theoretic framing in the body. This is intentional for a position paper—the title states the position provocatively, while the body develops the careful argument. The paper explicitly notes what would falsify the position (Section 10), which would be unnecessary if inevitability were meant literally.
>
> On the call-to-action: we will make concrete recommendations explicit in revision—(1) funding priority toward ML infrastructure (datasets, architectures, uncertainty quantification) over incremental traditional method development; (2) ML integration as a first-class capability in quantum chemistry software packages; (3) specific benchmark protocols including OOD generalization and reliability reporting, as the reviewer suggests. We appreciate the suggestion regarding ICML-style milestones.
>
> On data fidelity: the reviewer identifies a genuine constraint. Two developments address this. First, multi-fidelity approaches (discussed in Appendix B.7) combine abundant low-accuracy data with sparse high-accuracy calculations, allowing models to learn corrections across fidelity levels rather than being capped by any single reference method. Second, the data ceiling is rising—the Orbformer foundation model (Section 5.5) demonstrates that neural network wavefunctions themselves can generate high-fidelity labels with better scaling than traditional methods like CCSD(T), creating a virtuous cycle where ML both consumes and produces reference data.
>
> We will add the M-OFDFT reference (Zhang et al., 2024) alongside the STRUCTURES25 discussion in Section 7.1.

---

> > ### Author Rebuttal · Reviewer_eCaa · 2026-04-01
> >
> > The authors' responses have resolved most of my concerns. I will maintain my rating.

---

### Official Review · Reviewer_kPc4 · 2026-03-11

**Significance:** 3
**Argument Clarity:** 3
**Rating:** 3
**Confidence:** 3

**Questions:**

Question:
1. Please address my concerns in the "weakness" section.
2. How about the view "Both traditional methods and ML methods can work together to achieve good results." For example, recently, R2Scan datasets can improve the performance of ML potentials, and ML potentials learned from r2scan can scale to large structure [1].
3. For the first alternative view: "real chemistry may be tractable without ML.", the response is ML could be tractable. This is not a sound argument in my understanding. If we want to response to this view, we should try to prove that real chemistry is "intractable" without ML.

minor comment:
1. Parenthesis is next to the text without space in citations (e.g., materials science(Marzari (line 39-left))), there should be space between science and (Marzari. This issue persists throughout the paper.

[1] Kim, J., You, J., Park, Y., Lim, Y., Kang, Y., Kim, J., ... & Han, S. (2026). Optimizing cross-domain transfer for universal machine learning interatomic potentials. Nature Communications.

[2] Kuner, M. C., Kaplan, A. D., Persson, K. A., Asta, M., & Chrzan, D. C. (2025). MP-ALOE: an r2SCAN dataset for universal machine learning interatomic potentials. npj Computational Materials, 11(1), 352.

[3] Kaplan, A. D., Liu, R., Qi, J., Ko, T. W., Deng, B., Riebesell, J., ... & Ong, S. P. (2025). A foundational potential energy surface dataset for materials. arXiv preprint arXiv:2503.04070.

**Alternative Views Section:**

Yes

**Compliance With Llm Reviewing Policy A Conservative:**

Affirmed.

**Discussion Potential:**

2

**Final Justification:**

I increased the score because I might have been too strict for the evaluation of discussion potential. But overall, its potential is still somewhat limited. The field is now doing something close to what this position has stated and this the discussion potential score, at least in ICML.

**Paper Summary:**

This paper argues that machine learning (ML) the most promising path forward for solving quantum many-body problems quantum chemistry.  Five sub-positions are provided to justify this argument: 1) since complexity of the problem is not known, space-time trade-off via ML is the rational way to go. 2) traditional approaches show saturation performance and ML represents the most promising systematic framework for discovering specialized approximation. 3) ML can be interpretable with explainable AI techniques. 4) ML can go beyond "hand-crafting" method. 5) Quantum computer is useful, but AI/ML is still needed for a generalization engine. For these reasons, using ML is the most promising way to go beyond the performance saturation problem because of traditional methods.

**Position:**

Yes

**Position In Title:**

Yes

**Related Work:**

3

**Strengths And Weaknesses:**

Strengths
1. Easy-to-follow, and this can serve as a good introduction for anyone who wants to learn the situation of this field.
2. Arguments are reasonable for the perspective that ML is highly useful for the future understanding of quantum chemistry and is a highly scalable approach.


Weaknesses
1. This could be my misunderstanding, but what is the main position of this paper? Is it "we should stop developing traditional methods and let's do ML only because it is the most promising way?" or "ML is very useful?". According to writing guideline: "Introduction should state the position, using bold text". But there is no introduction section in the paper.
2. Lack of call-to-action or what the community should do differently from the current progress, where ML research in this field is now flourishing, as this paper also highlighted. At the same time, traditional methods are also being studied and improved, although some setbacks may have been observed.
3. As noted in conclusion in this paper as well, "What would falsify this position?" If the position is ML is the most promising way and non-ML should be disregard completely, perhaps it's still somewhat possible to debate. But if the position is "we should use ML". Given the current situation of the field, this might not stimulate further discussion beyond what is happening right now since the field is observing great progress on using ML and there are not much strong evidence to the best of my knowledge to deny using ML in this field.

**Support:**

3

---

> ### Author Rebuttal · Authors · 2026-03-29
>
> We thank the reviewer for their comments and address each concern below.
>
> **W1: Clarity of the main position.** We acknowledge the formatting issue—the paper lacks a labeled Introduction with a bold-text position statement as specified in the guidelines (and such changes will be made in revision). We clarify the central claim here: the position is not that "ML is very useful," but a stronger resource-allocation claim—that traditional quantum chemistry method development has reached diminishing returns, and the field should strategically redirect research priority toward ML-based approaches as the primary paradigm for discovering approximations, with traditional methods recast principally as generators of high-quality reference data. This is substantively different from the current status quo, where ML is treated as one tool among many while traditional method development continues to receive the majority of funding and researcher effort. The position is about **strategic priority, not elimination**. We will add a clearly labeled Introduction with a bold-text position statement in the revised manuscript.
>
> **W2: Call-to-action.** We agree this should be made explicit; it is currently implicit in the argument but never stated directly. The call-to-action is threefold: (1) funding agencies should prioritize ML infrastructure—datasets, architectures, uncertainty quantification—over incremental traditional method development; (2) quantum chemistry software packages should treat ML integration as a first-class capability rather than an afterthought, as the DM21 adoption gap (Appendix B.6) demonstrates that benchmark success without software integration is insufficient; (3) graduate training should emphasize ML methodology alongside traditional quantum chemistry, preparing researchers for the paradigm shift we argue is underway. We will make these concrete recommendations explicit in revision.
>
> **W3: Discussion potential.** We respectfully note that the discussion potential depends on the intended audience. While the ML community at ICML may find the pro-ML direction natural, the quantum chemistry community—which this paper aims to bridge toward—holds substantially different priors. The Medvedev et al. (2017) debate generated heated responses precisely because many practitioners remain deeply invested in traditional functional development. Our claim that this investment has reached diminishing returns and that traditional methods should be repositioned as data generators would be contested by a significant fraction of working computational chemists. **Position papers that bridge communities with divergent priors are where discussion is most needed.** The paper is also debatable within the ML community on whether neural scaling laws (Section 5.5) guarantee continued progress, or whether the adoption barriers in Appendix B.6 reflect deeper problems than engineering.
>
> **Q2: Traditional + ML synergy.** We agree, and this view is already central to our argument—not opposed to it. Section 6 (Position 5) frames traditional methods and quantum computers as reference data generators with ML as the generalization engine; Appendix B.7 discusses data quality and the complementary relationship explicitly. References [1–3] cited by the reviewer on r2SCAN datasets improving ML potentials illustrate precisely the paradigm shift we advocate: the most impactful role for traditional methods going forward is generating high-quality training data for ML rather than serving as the end-point computational tool. We will cite these works in revision.
>
> **Q3: "Real chemistry may be tractable without ML".** The reviewer raises a fair point. The response in Section 9.1 presents the decision-theoretic logic but does not consolidate the empirical evidence that supports it—this evidence is distributed across Sections 1.3, 2.3, and the Medvedev analysis, making the response to the objection read as weaker than it is. Fifty years of traditional effort have produced 400+ functionals without convergence toward universality; density accuracy has deteriorated post-2000 (Medvedev et al., 2017); strong correlation remains unsolved; no tractable analytical solution has emerged for broad chemical space despite sustained investment. This track record does not prove intractability of the chemically relevant subset—we explicitly acknowledge this remains unknown—but it constitutes strong empirical evidence that the subset does not admit simple analytical solutions, which is what makes the decision-theoretic case for ML compelling rather than merely abstract. In revision, we will consolidate this empirical case directly into Section 9.1 so that the response to the typical-case objection stands on its own.
>
> **Minor**: We will fix all citation formatting instances.

---

> > ### Author Rebuttal · Reviewer_kPc4 · 2026-04-03
> >
> > Thank you for the detailed rebuttal response.
> > I appreciate and have read the response.
> >
> > W1. Thank you. In my understanding the position is the boldtext is something along the line: "the field should strategically redirect research priority toward ML-based approaches as the primary paradigm for discovering approximations"
> >
> > W2, Q2, Q3, thank you for the clarification.
> > For W3, this response is reasonable.
> >
> > I will raise the score of discussion potential and my total score to 3.

---

### Official Review · Reviewer_7Qif · 2026-03-13

**Significance:** 3
**Argument Clarity:** 3
**Rating:** 5
**Confidence:** 4

**Questions:**

No additional questions.

**Alternative Views Section:**

Yes

**Compliance With Llm Reviewing Policy A Conservative:**

Affirmed.

**Discussion Potential:**

3

**Final Justification:**

Rebuttal has addressed my concern so I will keep my rate.

**Paper Summary:**

This paper argues that machine learning based approaches are the most promising future directions for quantum chemistry research. The position is supported by good space-time tradeoff, promising continued progress and good physical insight in machine learning approaches.

**Position:**

Yes

**Position In Title:**

Yes

**Related Work:**

3

**Strengths And Weaknesses:**

Strengths:
- This paper makes a valuable position towards future exploration direction in quantum chemistry. The proposed advantages of machine learning approaches would be insightful and enlightening for broad quantum chemistry researchers.
- This paper supports its proposed position with good evidences including literature reviews and detailed comparison between machine learning and traditional methods.
- The writing of this paper is generally good, clear and well-organized.

Weaknesses:
- One major weakness of machine learning approaches could be over-fitting and generalizability. For instance, a model fitted on small molecule set may perform poor on large molecules. Authors are suggested to discuss this point, showing if traditional methods or combining machine learning and traditional methods could overcome this weakness.

**Support:**

3

---

> ### Author Rebuttal · Authors · 2026-03-29
>
> We thank the reviewer for this observation on overfitting and generalizability. We agree this is a central challenge. The paper addresses it across several sections — out-of-distribution failure and uncertainty quantification in Section 8 and Appendix B.4, active learning in Section 5.1 (where models request new reference calculations when entering novel regions), and foundation models with system-specific fine-tuning in Section 5.3 — but we recognize these threads could be connected more explicitly to the overfitting concern the reviewer raises.

---

> > ### Author Rebuttal · Reviewer_7Qif · 2026-04-03
> >
> > My concerns have been addressed so I will keep my rate.

---

### Official Review · Reviewer_sMEn · 2026-03-16

**Significance:** 3
**Argument Clarity:** 2
**Rating:** 3
**Confidence:** 5

**Questions:**

In Section 1.2, the author claimed that traditional ab initio methods like FCI operate at one extreme of this tradeoff.  They are compact algorithms requiring minimal storage but must solve the problem from first principles for every new system, leading to intractable time complexity.  -> Just a little curious about why FCI requires minimal storage? From my opinion, as FCI needs all possible Slater determinants which might cost a lot of memory. Please feel free to correct me if I have misunderstood this point.

**Alternative Views Section:**

Yes

**Compliance With Llm Reviewing Policy A Conservative:**

Affirmed.

**Discussion Potential:**

3

**Paper Summary:**

This paper introduces current computational methods used in quantum chemistry, ranging from FCI to DFT. It also discusses recent machine learning approaches that aim to replace or augment traditional hand-crafted methods. In particular, the paper covers machine learning models for property prediction as well as wavefunction-based models designed to achieve higher accuracy.

**Position:**

Yes

**Position In Title:**

Yes

**Related Work:**

2

**Strengths And Weaknesses:**

In section 1.2, it said that Drug-like chemical space alone is estimated to contain 1033–1060 possible molecules(Bohacek et al., 1996; Reymond, 2015), yet modern machine learning potentials achieve useful accuracy with only 106–107 parameters(Batatia et al., 2023).
However, I am a little confused about how this paragraph relates to quantum machine learning. The discussion seems more closely connected to generative methods in machine learning for drug discovery rather than to quantum chemistry or quantum machine learning. It would be helpful if the authors could clarify the intended connection.

I also find Section 1.3 somewhat unclear. In particular, I did not see a clear connection between AlphaFold and the application of machine learning to quantum chemistry. While AlphaFold is an impressive demonstration of the power of machine learning, its success also heavily relies on the availability of large amounts of high-quality structural data. In many cases, machine learning primarily acts as a powerful function approximator whose performance strongly depends on the quality and scale of the training data. It would therefore be helpful if the authors could clarify how this example specifically motivates or relates to the context of quantum machine learning.


Section 2 provides a good introduction to widely used quantum chemistry algorithms and clearly demonstrates the current limitations of these methods. This serves as a strong motivation for introducing machine learning–based approaches.

Section 3 does not appear to be sufficiently well justified. The motivation for the large number of explainability methods in machine learning largely stems from the fact that modern ML models are often difficult for humans to interpret. Methods from explainable AI (XAI), such as SHAP, can only partially reveal how a model arrives at its decisions. However, it is not clear how these techniques are directly applicable to machine learning models used in quantum chemistry. In particular, SHAP was originally developed primarily for classification settings. When ML models are used to predict physical quantities such as energies, the problem becomes a regression task, where the applicability and interpretability of SHAP are less straightforward. Moreover, the black-box nature of many ML models can further complicate the interpretation of the learned physical relationships. Additionally, the discussion of DM21 appears somewhat disconnected from the topic of explainability. While DM21 addresses the fractional electron problem in traditional DFT methods, I did not clearly see how this example relates to the explainability of machine learning models. It would be helpful if the authors could clarify these connections.

Section 4 is a strong part of the paper. In recent years, multiple machine learning models, particularly equivariant neural networks, have been developed to address these challenges and have achieved significant success. However, the related work could be further expanded. For example, important models such as Allegro and eSCN could also be included in the discussion.

In Section 5, the introduction of FermiNet and PauliNet is informative and provides useful background on wavefunction-based machine learning approaches. The claim that “This suggests the path to higher accuracy is clear: invest in larger models rather than hoping for conceptual breakthroughs in ansatz design.". However, it may also be helpful to discuss some of the challenges associated with these methods. In particular, the optimization of such models can be difficult and often requires substantial GPU resources even for single molecular structures. Highlighting these limitations could provide a more balanced perspective and further motivate continued research in this area.

**Support:**

2

---

> ### Author Rebuttal · Authors · 2026-03-29
>
> We thank the reviewer for their detailed and technically sharp comments. We address each concern below.
>
> Section 1.2: Chemical space compression and drug discovery. The reviewer is right to note that the 10³³–10⁶⁰ estimates pertain specifically to drug-like chemical space. This example is not intended as a claim about drug discovery per se; it serves as the best-quantified illustration of a general principle—that ML can represent vast chemical spaces with modest parameters (10⁶–10⁷), achieving compression ratios of 10²⁶–10⁵³. Drug-like space provides concrete numbers because it has been studied most extensively; the same compression principle applies to the broader spaces relevant to quantum chemistry (materials, catalysis, extended systems). We will clarify this intended scope in revision.
>
> Section 1.3: AlphaFold analogy. The analogy is developed along several specific axes: NP-hard general problem with a structured subset constrained by physical selection; fifty years of sustained effort by the community; resolution through learned specialization (93M parameters trained on ~170K structures); no simple algorithm or physical principle subsequently emerging to replicate the result. These parallels are structural, not superficial. The reviewer's concern about data availability is a valid point—AlphaFold relied on large-scale experimental data. We note that Section 4 addresses this directly: high-throughput quantum chemistry databases (QM9: 134K molecules; Materials Project: hundreds of thousands of materials; MPtrj: 1.6M configurations spanning 89 elements) provide the analogous training signal. We will make this cross-reference explicit in revision.
>
> Section 3: Interpretability, SHAP, and DM21. The reviewer identifies a genuine structural issue. Section 3 conflates two distinct arguments: (1) that XAI tools provide a developing—not yet mature—toolkit for post-hoc interpretation of ML models in chemistry, and (2) that architecture design itself constitutes a form of physical insight, which is the deeper claim. SHAP is mentioned as an example of the direction the field is moving, not as a specific proposal for quantum chemistry energy prediction; we will clarify this distinction and note the regression-vs-classification nuance the reviewer raises. DM21 belongs to the second argument—its architecture enforces fractional electron constraints, and its success on systems where conventional functionals fail validates that these constraints correspond to genuine physical requirements. The insight arises from the architecture choice, not from post-hoc explanation. We will add a clarifying sentence at the transition between Sections 3.1 and 3.2 to distinguish post-hoc interpretation from architecture-as-hypothesis.
>
> Section 4: Related work. We agree that the discussion of equivariant architectures should be expanded. We will include Allegro and eSCN alongside the currently cited NequIP, MACE, and PaiNN in revision.
>
> Section 5: Challenges of neural network wavefunction methods. The reviewer makes a fair point. Section 8 acknowledges computational challenges broadly, but the specific difficulty of optimizing FermiNet-type models—substantial GPU resources even for single molecules—deserves explicit mention in the context of Section 5's scaling discussion. We will add this qualification in revision.
>
> FCI and storage. The reviewer's technical point about FCI memory requirements is correct—FCI requires enormous memory for the CI vector, and "minimal storage" is imprecise as written. Our intended meaning operates within the P/poly framework established earlier in the section: "storage" refers to stored specialization—precomputed domain knowledge, the "advice" in non-uniform computation—not computational intermediates. FCI stores no such specialization; it is a uniform algorithm that recomputes everything from first principles for each new system. The reviewer's observation actually strengthens our argument: FCI is expensive in both time (factorial scaling) and space (all Slater determinants), while ML trades precomputation time (training) for cheap inference with stored specialization (weights). Indeed, traditional quantum chemistry already performs this tradeoff at smaller scale—storing precomputed basis sets, contracted integrals, and optimized active space selections rather than recomputing them from scratch—which suggests the community has long recognized the value of trading space for efficiency. ML extends this principle to a qualitatively larger scale. We will revise the language to make this distinction precise.

---

> > ### Author Rebuttal · Reviewer_sMEn · 2026-04-02
> >
> > Thank you for the reviewer’s detailed feedback. Look forward to seeing the strengthened and revised version in the future.

---

### Decision · Program_Chairs · 2026-04-30

**Decision:**

Accept (regular)

**Comment:**

This paper received mixed reviews (2 borderline reject, 2 accept).

There is general appreciation for the position the paper takes in quantum chemistry, clarity of exposition and presentation, high educational / survey value, and technically rigorous arguments. On the other hand, there were concerns raised around whether the stated position is indeed debate-worthy, and scalability of ML for specific quantum chemistry problems.

The authors addressed the reviewer concerns, after which one of the reviewers raised their rating. Although the final reviews remain mixed, on balance, an accept decision was reached.